# Exosomes: A New Hope for Angiogenesis-Mediated Bone Regeneration

**DOI:** 10.3390/ijms25105204

**Published:** 2024-05-10

**Authors:** Martin Lange, Patrick Babczyk, Edda Tobiasch

**Affiliations:** 1Cardiovascular Research Center and Department of Cellular and Molecular Physiology, Yale University School of Medicine, New Haven, CT 06510, USA; 2Department of Natural Sciences, University Bonn-Rhein-Sieg, D-53559 Rheinbach, Germany

**Keywords:** angiogenesis, bone development, bone regeneration, exosome biogenesis, extracellular vesicles, mesenchymal stem cells, endothelial cells, vascular development, exosome patents, clinical studies

## Abstract

Bone is a metabolically dynamic structure that is generally remodeled throughout the lifetime of an individual but often causes problems with increasing age. A key player for bone development and homeostasis, but also under pathological conditions, is the bone vasculature. This complex system of arteries, veins, and capillaries forms distinct structures where each subset of endothelial cells has important functions. Starting with the basic process of angiogenesis and bone-specific blood vessel formation, coupled with initial bone formation, the importance of different vascular structures is highlighted with respect to how these structures are maintained or changed during homeostasis, aging, and pathological conditions. After exemplifying the current knowledge on bone vasculature, this review will move on to exosomes, a novel hotspot of scientific research. Exosomes will be introduced starting from their discovery via current isolation procedures and state-of-the-art characterization to their role in bone vascular development, homeostasis, and bone regeneration and repair while summarizing the underlying signal transduction pathways. With respect to their role in these processes, especially mesenchymal stem cell-derived extracellular vesicles are of interest, which leads to a discussion on patented applications and an update on ongoing clinical trials. Taken together, this review provides an overview of bone vasculature and bone regeneration, with a major focus on how exosomes influence this intricate system, as they might be useful for therapeutic purposes in the near future.

## 1. Introduction

### 1.1. Angiogenesis

The cardiovascular system consists of the heart, lymphatic vasculature, and blood vessels. Its major function is to supply organs, tissues, and cells with nutrients and oxygen, as well as the transport of waste products, signaling molecules, and immune cells [1,2].

Blood vessels can form via vasculogenesis or angiogenesis. Vasculogenesis is the de novo formation of embryonic blood vessels. During this process, endothelial progenitor cells, known as angioblasts, specify and migrate to assemble the dorsal aorta and cardinal vein [2,3]. Afterward, the vascular network further expands via angiogenesis [4]. This describes the process of endothelial cell (EC) migration, proliferation, and sprouting from preexisting blood vessels. First, pericytes detach from the blood vessel to allow the initiation of sprouting [5]. During sprouting, growth factors induce the specification of a tip cell that forms filopodia, which migrate and guide the sprout into an avascular area. This invasive tip cell is followed by stalk cells, which possess a high proliferative capacity [6,7]. Tip cells emerge from venous origin. After tip cell specification, the tip cell migrates and connects to a neighboring artery [8]. This process is mediated by CXCL12 and its receptor, CXCR4. These findings were further supported by lineage-tracing experiments in mice [8]. The role of CXCR4 and its crosstalk with the Notch pathway were further investigated. It was shown that Notch activity in tip cells is required for the initiation of CXCR4 expression [9,10]. Afterward, the Notch signaling in the tip cell is required to reduce CXCR4 expression, preventing excessive blood vessel growth [9,10]. More recent work in the mouse retina identified different types of tip cells (S-tip cells and D-tip cells) [11]. These distinct tip cells show differential expression of genes related to the blood–retina barrier, metabolic requirements, and extracellular matrix components [11]. A similar mechanism of different tip cells was observed during zebrafish hindbrain vascular development. Here, two separate modes of sprouting emerge from the same venous origin to form first the basilar artery and then the central arteries [12,13]. This process was shown to depend on different vascular endothelial growth factor A (VEGFA) ligands [12,13]. After the formation and stabilization of endothelial cell–cell contacts, the newly generated blood vessel forms a lumen and matures with the arrival of mural cells [14] (Figure 1). 

### 1.2. Bone Specific Vascular Development

The bone vasculature is mainly formed by angiogenesis. The vascular invasion in murine long bones starts at around embryonic day (E) 13 to 14 and reaches completion in young adult animals [15]. This invasion is an important step in the process of osteogenesis. It is initiated by the extracellular matrix and growth factor signaling. Considered to be the master regulator of angiogenesis, VEGFA and its receptor VEGF receptor 2 (VEGFR2 or KDR) interact to induce sprouting, migration, and proliferation of ECs [12,16]. Notably, VEGFR2 is also expressed by other bone-resident cells, such as osteoblasts and osteoprogenitor cells, pinpointing important implications for osteogenesis and bone repair [17,18]. Osteogenic progenitors and hypertrophic chondrocytes secrete VEGF to stimulate angiogenesis [19,20,21,22]. Angiogenesis in the bone is coupled to ossification. First, endothelial cells extend protrusions from the periosteal vasculature to form a vessel plexus in embryonic femoral cartilage. This results in the primary ossification center (POC) and later an epiphysial secondary ossification center (SOC) [18,23,24,25] (Figure 2). During the extension of the POC, blunt vessel buds are formed and extend from the vascular loops located close to the hypertrophic chondrocytes in the growth plate [26,27]. Those distal vessel buds are fully lumenized and are formed by several ECs. The buds continue to extend into empty space generated by apoptotic chondrocytes and form new vessel loops by anastomosis of adjacent buds [26,28,29,30] (Figure 2).

### 1.3. Vascular Homeostasis in Bone

Endothelial cells of different organs must fulfill a variety of tissue-specific functions, such as maintenance of the blood–brain barrier (BBB), support of metabolic processes in the liver, gas exchange in the lung, or blood ultrafiltration in the kidneys [31,32,33,34,35,36,37,38]. This specificity of endothelial cells is achieved by the formation of specialized morphological structures and their underlying gene expression patterns [39,40,41,42]. 

In long bones, ECs interact with osteoprogenitors during bone formation and healing processes [15,27,43]. In addition, they provide a microenvironment, known as a niche, for hematopoietic stem cells (HSCs). This niche is required for blood formation and has further implications for a variety of hematological diseases [44,45,46]. As a result of the above-described developmental processes, the bone vasculature is made up of three distinct structures known as buds, arches, and columns. The arches are connected at their distal side with column-like capillaries in the diaphysis [43]. These column-shaped blood vessels are associated with osteoprogenitor cells and bone mesenchymal stromal cells (MSCs). The population of MSCs possesses a large variety of functions and influences hematopoiesis and osteogenesis, which are described in detail in other reviews [47,48,49,50,51,52]. The distinct vascular structures found in bones have long been described [28,29,30]. However, more recent studies shed more light on the cellular heterogeneity of bone capillary ECs. The identified EC subtypes present unique molecular identities with specialized functional roles [27,43]. The expression of two cell surface markers, the sialoglycoprotein endomucin (EMCN) and the cell adhesion molecule CD31, can be used to differentiate between three different EC populations. The metaphysial ECs present in vessel buds and the resulting columns express high levels of CD31 and EMCN. This differentiates these cells from sinusoidal ECs in the diaphysis, which express lower levels of EMCN and CD31. These two cell populations are described as Type H in the diaphysis (buds and columns) and Type L in the diaphysis (sinusoidal), respectively [43]. The third endothelial population, identified in long bones during embryonic and early postnatal development, was named type E (embryonic) ECs [53]. Type E ECs are characterized by high expression of EMCN and CD31. However, this subpopulation is unique due to a higher expression of CD31 and a lower expression of EMCN compared to type H ECs [53]. Type E ECs can develop into type L sinusoidal cells as well as into type H ECs in postnatal stages [53]. 

The demands of bone tissue vary from development to adulthood. The proportion of the different EC populations in bone varies according to the live stage. During embryonic development, type E ECs present the majority of ECs. While some L-type ECs are present at this stage, only a small number of H-type ECs can be found. After birth, the proportion of type E ECs decreases, while the number of type L ECs constantly increases throughout life. The type H ECs reach their highest quantity around P6 and will afterward decline during adulthood and in aged mice [53]. Interestingly, the bone EC identity is less static than assumed. Cell tracing experiments using a tamoxifen-inducible Apln-CreERT reporter revealed the plasticity of bone EC subpopulations. For example, type E ECs can give rise to type H ECs, and both cell types (type E and H) can change into type L and arterial ECs during postnatal development [53,54]. 

### 1.4. Bone Vascular Homeostasis under Pathological Conditions

As described above, the vasculature of bone changes in response to developmental stages and aging and is linked to bone formation during development. However, the bone vasculature plays a crucial role in the context of pathological conditions as well. Bone remodeling is a continuous process throughout life. This can be observed during the constant replacement of older bone tissue as well as the repair of microfractures. The coordinated interplay of osteoclasts (that dissolve bone) and osteoblasts (that produce new bone) is crucial for the maintenance of homeostasis. The imbalance of these two cell types, because of dysregulation or dysfunction, results in a range of pathological conditions [55]. Also, bone density progressively decreases with increasing age [56,57,58]. However, patients suffering from osteoporosis show a dramatic increase in bone density loss. Osteoporosis is a skeletal disorder that correlates with increased age and can occur in women as well as men. However, women are more likely to develop osteoporosis, especially after menopause. In general, osteoporosis is known to compromise bone strength and predispose the patient to an increased risk of fracture [57,59,60]. 

During aging, type H vessels are continuously lost, which correlates with decreased osteogenesis, fracture healing, and overall bone quality [43,61,62]. This process can be counteracted by endothelial-specific deletion of the von Hippel–Lindau (VHL) protein, which results in increased activity of the transcription factor HIF and its target genes. As a result, type H vessels, osteoprogenitor cells, and thus overall trabecular bone formation were increased [43,63]. Similarly, pharmacological stabilization of HIF using deferoxamine mesylate (DFM) also resulted in increased bone density compared to age-matched controls [46,64]. Another pathway reported to be important in the coupling of bone formation and angiogenesis is the Dll4-Notch pathway. In contrast to ECs in other organs, Notch signaling promotes EC proliferation and vessel growth in postnatal bones [27]. Endothelial-specific disruption of the Notch pathway resulted in impaired blood vessel growth and morphology, while at the same time also impacting the size of bones and overall bone quality [27]. This was found to be a result of the missing angiocrine release of Noggin, which usually acts as an antagonist of BMPs [27,65]. Administration of recombinant Noggin restored bone quality in endothelial-specific Notch pathway mutants [27]. Following these findings, an artificial Notch ligand with a high affinity for bone tissue was generated and applied to mice. As a result, bone formation, specifically in male mice, was successfully induced. Interestingly, the observed effects were due to Notch activation in mesenchymal stem cells in the bone [66]. Human osteoporosis patients show a similar decrease in H-type blood vessels in the bone, highlighting the importance of these findings [67]. 

Patients with osteoporosis are more likely to suffer from bone fractures. This traumatic event is the most common large-organ injury in otherwise healthy humans. While bone is one of the few tissues in adults that can completely regenerate, fracture healing is a complex, multistep process [68,69]. As one of the first steps, a hematoma is formed around the fractured region of the bone, followed by an invasion of inflammatory cells that help to form a fibrin clot to stop bleeding [69,70,71]. Bone formation and repair following injury can occur in two ways. Unstable and hypoxic fractures are usually repaired via endochondral ossification. In contrast, stable bone fractures have sufficient oxygen and nutrient supply to allow direct differentiation of mesenchymal cells into osteoblasts, as seen in intramembranous ossification during developmental processes [72,73]. Blood vessels from three different tissues have been implicated in the restoration of blood circulation after bone injury. These tissues are bone marrow, compact bone, and periosteum, which cover the outer surface of bones [74,75]. High levels of VEGF A within the hematoma promote the ingrowth of blood vessels, which is crucial for the formation of a soft callus at the healing area. This soft callus is formed by chondroblasts and osteoblasts that promote bone and cartilage formation. After further maturation, a hard callus is formed, which will finally be remodeled into mature bone [72,75,76]. The importance of vascular invasion for callus formation and bone healing is supported by many studies. The application of the pro-angiogenic factor VEGF A to fractured areas has the potential to improve the vascularization process and, thus, the healing process [77]. The opposite was achieved using a soluble form of VEGFR1, which acts as a decoy receptor for VEGF A and thus reduces angiogenic sprouting. Consequently, blood vessel formation and callus mineralization were reduced, and overall fracture healing was impaired [78,79,80]. 

Several other growth factors and their pathways, such as fibroblast growth factor (FGF) or transforming growth factor β (TGFβ), have been shown to be involved in bone formation and fracture healing [81,82,83,84]. The expression of FGF and its receptor was shown to be increased at sites of fracture healing [85]. Furthermore, the application of FGF contained in absorbable collagen sponges was shown to stimulate bone vascularization and osteogenesis during bone repair [86]. TGFβ signaling is known to be involved in bone formation and repair. Similar to FGF, TGFβ and the receptor Tβ-RII showed increased expression in rats after osteotomy [87]. Local as well as systemic injections of TGFβ resulted in improved callus formation and overall bone health [88,89]. The observed effects seem to be mainly mediated by chondroblasts and osteoprogenitor differentiation during the healing process of bone fracture [82,90]. In a recent clinical trial, a novel orally administered parathyroid hormone tablet was tested in postmenopausal women with low bone mineral density or osteoporosis. After 6 months of treatment, the authors reported no drug-related serious adverse events, while the tablets appeared to increase bone mineral density due to the dual mechanism of stimulating bone formation and inhibiting bone resorption [91]. 

### 1.5. Exosomes: Definition, Discovery, Classification, Isolation, and Characterization

Exosomes are a subtype of extracellular vesicles (EVs), which are defined as a spheroid structure composed of a lipid bilayer with various cargo or contents. They are not able to replicate by themselves but are released from eukaryotic and prokaryotic cells and involved in intercellular communication. 

EVs were first reported by Chargaff and West in 1946, when they observed procoagulant platelet-derived particles in blood plasma and called it “platelet dust” [92,93]. Almost 40 years later, two research groups reported the release of vesicles generated after the formation of multi-vesicular bodies in reticulocytes [94,95]. EVs have also been found in other biological fluids, like salvia, semen, urine, and breast milk [96]. 

Today, EVs are classified into four types according to their size: (I) endosomal exosomes (50–100 nm); (II) microvesicles (MVs) (20–1000 nm); (III) membrane particles (50–600 nm); and (IV) apoptotic vesicles (1000–5000 nm) [97]. Since some groups only distinguish two types of EVs (exosomes and ectosomes), the International Society for Extracellular Vesicles stated the following in the “Minimal Information for Studies of Extracellular Vesicles 2018,” with respect to EV nomenclature: “EV is the preferred generic term for the subject of our investigations, and subtypes should be defined by physical and biochemical characteristics and/or conditions/sources. When other terms are used, careful definition is required” [98]. For more details on the classification of different extracellular vesicles, see Table 1.

The standard isolation procedure of EVs from cell supernatant is a multistep centrifugation procedure consisting of several steps, starting from 300 g (to remove cells and debris), over 10,000 g (to remove proteins), and up to 100,000 g (ultracentrifugation). Over the past years, several additional methods have been developed, such as density gradient ultracentrifugation to isolate specific populations, antibody-based techniques like capture beads in magnetic-activated cell sorting, precipitation using polymers, or size-exclusion chromatography [99,100,101,102,103]. For a comparison of the different isolation techniques, see Table 2.

To determine which type of EV is isolated, its biochemical properties and biological functions have to be defined. This can be achieved by determining size, shape, content, and surface markers. Methods for these characterizations are atomic force microscopy (AFM) or transmission electron microscopy (TEM) both for visualization and characterization of their structure, morphology, and size; nanoparticle tracking analyses (NTA) or tunable resistive pulse sensing (TRPS) for determination of the size and concentration of particles; polymerase chain reaction (PCR), microarray and second generation sequencing (SGS), and third-generation sequencing (TGS) for their content; Western blotting and flow cytometry for the characterization of surface markers; and (xi) fixation for in situ imaging [99,104,105,106]. For a more detailed comparison of these methods, see Table 3.

### 1.6. Biogenesis of Exosomes, Cargo, and Characteristics

The term exosomes was coined by R.M. Johnstone [107]. The biogenesis of exosomes is based on the exocytosis of multivesicular endosomes, also called multivesicular bodies (MVB). This process is divided into three stages. The first stage is the formation of endocytic vesicles from pits in the plasma membrane. The second stage is the formation of MVBs by inward budding of endosomal membranes. During this stage, MVBs are loaded with their cargo. Within the third stage, these MVBs can then be degraded by the lysosome or fuse with the membrane of the cell and thus be released as exosomes (Figure 3) [108]. The best-known mechanism is carried out by the endosomal sorting complex required for transport (ESCRT). The ESCRT is composed of four complexes: ESCRT-0, -I, -II, and -III, with associated proteins. The ESCRT-0 complex recognizes ubiquitinated proteins in the endosomal membrane. ESCRT-I and -II complexes are responsible for membrane deformation into buds with sequestered cargos. ESCRT-III drives vesicle scission [109]. In addition to ESCRT-dependent pathways, other ESCRT-independent mechanisms for EV biogenesis have been described. They involve the hydrolysis of sphingomyelin into ceramide or proteins like tetraspanins and CD63 [108,110]. Tetraspanins are also involved in the cargo secretion of EV and its uptake by recipient cells [110]. 

The contents of the MVB are often degraded by hydrolases if the former merge with lysosomes. However, in some cases, MVB may fuse with the plasma membrane. That allows the release of their contents into the extracellular environment. Specific MVB features include the presence of tetraspanins, besides other molecules generally present in the late endosomes [e.g., major histocompatibility complex (MHC) class II, in antigen-presenting cells] [108,111].

The cargo of EVs and, thus, exosomes depends on the donor cell type and their physiological conditions. They show a specific differential selection of proteins when generating such vesicles. The main content found in EVs includes proteins from the endosome itself, plasma membrane, and cytosol. Proteins from the nucleus, mitochondria, endoplasmic reticulum, and Golgi complex are usually absent in EVs. In addition to proteins, lipids, and nucleic acids, especially messenger RNA (mRNA), microRNAs (miRNA), and non-coding RNA (ncRNA), they can be found in high numbers [108,112,113].

The lipid composition of EVs depends on the origin cell type. The lipid bilayer mainly contains components from the plasma membrane (Figure 3). They may be enriched with other proteins, including phosphatidylserine, desaturated phosphatidylethanolamine, desaturated phosphatidylcholine, sphingomyelin, GM3 ganglioside, and cholesterol [104].

Some biomarkers, like tumor susceptibility gene 101 (TSG101), charged multivesicular body protein 2a (CHMP2A), and Ras-related protein Rab-11B (RAB11B), in association with CD9, CD63, and CD81 proteins, are used for exosome characterization. Comparative analyses of nucleic acids between the cells and EVs may show differential contents [99,114,115,116].

## 2. Exosomes in Bone Vascular Development and Homeostasis

After inoculating patients with MSCs to promote tissue regeneration, it was shown that less than 1% of these cells were left in the damaged tissue after one week [117,118]. Nevertheless, the strategy produced positive results in tissue regeneration and functionality [119]. Thus, it was postulated that the regenerative effect of MSCs was not primarily due to their capacity to proliferate and differentiate into the specific cell types of the damaged tissue but that their main function might derive from their paracrine actions through the production of different factors [117,120,121]. This hypothesis is supported by studies using conditioned media from MSC cultures, resulting in a regenerative capacity that can be higher than that of MSCs themselves [122,123]. These results demonstrate the therapeutic relevance of the MSC secretome.

The MSC secretome has a fraction composed of soluble factors, metabolites, and other encapsulated microvesicles to which exosomes belong. Kusuma and colleagues could show that the latter is mainly responsible for the therapeutic properties of conditioned media from MSC cultures [124]. EVs can regulate different physiological processes like proliferation, differentiation, and migration [125,126]. The therapeutic features of MSC-derived EVs are mainly due to their immunomodulatory properties. 

The use of exosomes in therapy has significant advantages if compared to complete MSCs [127]. First, they can be isolated and stored at low temperatures (−80 °C) until needed. Second, their content is encapsulated, protected from degradation in vivo, and thus relatively stable. Third, exosomes have a reduced risk of undesirable side-effects, like immune rejection, cell dedifferentiation, or tumor formation, which can arise after applying some exogenous cells [118,128,129,130].

### 2.1. Exosomes in Angiogenesis

As mentioned before, angiogenesis is the development of new blood vessels from existing capillaries or capillary veins. Under normal circumstances, angiogenesis is in a state of equilibrium. Once this equilibrium is disturbed, the vascular system is activated to cause degeneration of blood vessels by overgrowth or suppression of the vascular system [131,132,133,134]. Angiogenesis is a complex process coordinated by pro-angiogenic and anti-angiogenic factors. These processes include the degradation of the basement membrane during activation, proliferation, and migration of endothelium. This results in the formation of new blood vessels and vascular networks. These complex processes require different molecules from a variety of cells [135,136,137,138]. 

In numerous studies, the angiogenic activity of MSC-derived exosomes has been investigated with regard to their impact on different cell types, especially endothelial cells. HUVECs have been treated with exosomes obtained from MSCs, derived from adipose tissue (AT-MSC) by Ren and bone marrow-derived MSC (BM-MSC) by Shabbir. The results showed that endothelial cells engulfed the exosomes, increasing proliferation and migration and enhancing their angiogenic capacity [139,140]. 

Many studies focusing on angiogenesis investigated wound healing models. When conducting these experiments with respect to signaling molecules and their activated pathways in endothelial cells, it was reported that BMSC-derived exosomes can activate the VEGF and Hippo pathways by regulating cell-to-cell contact and actin cytoskeleton dynamics [141]. The pro-angiogenic effect of MSC-derived exosomes has been observed by several groups. The results showed improved or enhanced angiogenesis via hypoxia-treated donor cells or exosome cargos like miRNA [142,143,144]. In line with this, Ning and colleagues showed that miRNA-153-3p reduced EC apoptosis and improved angiogenesis [145]. Similarly, Pan and colleagues showed that miRNA-126 from MSCs enhanced angiogenesis via the PI3K/AKT/eNOS pathway [146]. Several chemicals, physical conditions, and scaffolds were tested for their pro-angiogenic properties. This so-called preconditioning can be achieved, e.g., by inducing hypoxia or chemical compounds. Hypoxia is a well-known inducer of angiogenesis and can be achieved by culturing cells with deferoxamine (DFO) [147,148]. ECs treated with DFO showed increased proliferation, migration, and angiogenesis. The Liang group used low doses of dimethyloxaloylglycine (DMOG) in BM-MSC cultures. These preconditioned exosomes activated the AKT/mTOR pathway to stimulate angiogenesis in HUVEC [149]. The PI3K/AKT signaling pathway could also be activated with exosomes containing miR-126 [150]. Yu and colleagues treated BM-MSCs with atorvastatin and showed that these exosomes (ATV-exos) promoted proliferation, migration, and tube formation and increased VEGF expression in HUVECs. Additionally, they found an upregulation of miR-221-3p after ATV-exo stimulation [151].

Another way to enhance angiogenesis via exosomes is to use genetic engineering, or bioengineering [152,153]. Overexpression of specific proteins, or miRNAs, resulted in proangiogenic effects in ECs. For example, overexpression of islet-1 (ISL1) enhanced the paracrine effect of MSCs and promoted angiogenesis in a myocardial infarction model [143]. On the other hand, Chen and colleagues transfected MSCs with an miR-150-5p expression plasmid and showed decreased migration and tube formation in HUVEC [154]. Ma and colleagues generated small extracellular vesicles loaded with mRNA encoding VEGFA and BMP-2, which were loaded onto an injectable hydrogel for bone regeneration in rats with femur critical-size defects. This resulted in enhanced angiogenic–osteogenic activity and overall improved bone regeneration [155].

In addition to preconditioning and bioengineering, physical conditions may have an impact on exosomes. Gao cultured MSCs in 2D or 3D and analyzed HUVECs after exosome treatment. The cells treated with 3D-exos showed enhanced proliferation, migration, tube formation, and in vivo angiogenesis [156]. In a more physiological context, it was shown that tensile stretch applied to the bone, known as the Ilizarov treatment, resulted in the formation of a metaphysis-like architecture composed of type H endothelial cells. Furthermore, the authors observed that tensile stretch-stimulated bone marrow endothelial cells secreted exosomes enriched with vital molecules, which could promote segmental bone defect healing [157]. 

For use in regenerative medicine, e.g., bone regeneration, cells or exosomes can be loaded on gels or scaffolds [158,159,160,161,162,163,164]. These strategies will be discussed in the following chapter. A summary of MSC-derived exosomes and their impact is illustrated in Figure 4.

### 2.2. MSC Exosomes in Bone Angiogenesis and Vascular Development

The formation of blood vessels during bone formation is crucial in regenerative medicine. Newly formed bone has to be supplied with oxygen and nutrients. Several approaches have been made to combine osteogenesis and angiogenesis. These studies usually focused on the differentiation process of MSCs towards osteoblasts and the impact on endothelial cells regarding proliferation, migration, and tube formation in vitro. Also, in vivo studies applying exosomes from different sources showed improved osteogenesis and angiogenesis in animal models and patients [165]. The strategy of using scaffolds and exosomes for this purpose was pursued by several groups. Exosomes from rat BM-MSCs were combined with a hydrogel (PG/TCP) to investigate the effect on osteogenesis and angiogenesis by Zhang in 2021 [165]. More recently, an electrospun scaffold was used in combination with M2 macrophage-derived exosomes. During in vitro experiments, these exosomes increased cell migration, tube formation, osteogenic differentiation, and anti-inflammatory macrophage polarization. The authors observed enhanced vascularized bone formation after applying an exosome-loaded nanofibrous scaffold to a critical-sized rat cranial bone defect model [166]. Wang and colleagues compared exosomes from M1 and M2 macrophages with respect to their pro-angiogenic properties. They could show that, compared to M1-Exos, M2-Exos showed a higher osteogenic and angiogenic potential [167]. The group of Fang showed that exosomes from human umbilical cord MSCs (hUC-MSC) promoted angiogenesis and osteogenesis via miR-21-5p [168]. In addition, exosomes from hUC-MSCs were used by Zhang and colleagues for bone repair experiments. They embedded exosomes in a hyaluronic acid hydrogel and combined it with a nanohydroxyapatite/poly-ε-caprolactone scaffold (nHP) to repair cranial defects in rats. The study showed enhanced bone regeneration, and in vitro experiments demonstrated improved proliferation, migration, and angiogenic differentiation of endothelial progenitor cells (EPC). The authors stated that miR-21 was the potential intercellular messenger that promoted angiogenesis by upregulating the DLL4/NOTCH pathway [169]. Another group of preconditioned exosomes collected from adipose tissue-derived stem cells, which were also enriched in miR-21-5p, reproduced the proangiogenic effects observed in HUVECs. The exosomes were used in a mouse osteoporotic fracture model, which improved new bone formation and bone mineral density compared to control mice [170]. The use of scaffolds to enhance the two coupled events of osteogenesis and angiogenesis during bone formation was also investigated using pre-treated exosomes. As already described above, hypoxia-induced exosomes enhance angiogenesis. Liu and colleagues applied hypoxia-induced MSC–exosomes to investigate bone fracture healing and showed that administration of these exosomes promoted angiogenesis through miR-126 and the SPRED/Ras/Erk pathway [171]. Hypoxic pre-conditioning of human exfoliated deciduous teeth (SHED) also resulted in enhanced angiogenesis for bone repair. Gao and colleagues used PDA scaffolds additionally and showed improved bone repair through vascularization [156]. Deng and colleagues showed that hypoxia-preconditioned bone marrow MSCs secreted more biglycan-rich extracellular vesicles, which promoted the proliferation, migration, differentiation, and mineralization of osteoblasts [172]. Another group used exosomes derived from endometrial mesenchymal stem cells in combination with a porous bioactive glass scaffold. They applied these scaffolds to critical-size bone defects in rats and observed improved osteogenesis, which was assessed by microCT and histological staining [173]. 

The group of Kobayashi showed that MSC-derived exosomes promote bone regeneration during the early stages as well as enhanced angiogenesis [174]. Wang and Xu modified MSCs to generate TGFß1-carrying exosomes and showed the maintenance of bone architecture in a cartilage damage model. They also showed an inhibition of osteoclastogenesis by suppressing the MAPK pathway in vitro via miR-135b [175]. A summary of the key players and the crosstalk between angiogenesis and osteogenesis is displayed in Figure 5.

During bone development, the formation of blood vessels within the bone is crucial, and exosomes seem to play a key role in this process. Different studies were conducted investigating the impact of exosomes on bone vascular development. One strategy is the use of disease models like vascular dementia (VaD), diabetic bone defects, or osteoporosis. Han and colleagues showed that miR-154-5p inhibition in BM-EPCs improved angiogenic parameters in a VaD model [176]. The group of Song improved the healing of diabetic bone defects via an exosome delivery system [177]. Another group applied an MSC-derived exosome delivery system to an osteoporosis model [178]. Hypoxia models display another interesting model for bone vascular development. Hypoxic preconditioned exosomes (SHED) from donor cells enhanced angiogenesis in rat calvarial defects, and hypoxic preconditioned BM-MSC-released exosomes improved HUVEC angiogenesis [179,180]. Overall, exosomes released by MSCs, or chondrocytes, seem to have an impact on vascular development, which is often mediated by miRNAs or lncRNAs [181,182,183].

## 3. Patents and Clinical Studies

### 3.1. Patents: Exosomes in Bone Regeneration

In the following section, patents involving the production or application of exosomes in the context of bone regeneration will be discussed (see Table 4). 

One of the first steps to using exosomes for therapeutic purposes is to stimulate the donor cell to produce enough exosomes. The following patent describes a method for producing exosomes from cells by electrical stimulation. More specifically, mammalian cells are cultured, and radiowave electrical stimulation (0.05 to 3 MHz) is applied to improve exosome secretion and functionality (WO2020256520A1).

Another patent focusing on large-scale production of exosomes for clinical use describes the culturing of mesenchymal stromal cells in the presence of IFNγ, TNFα, IL-1β, and IL-17 to prime the cells. It further describes the use of an automated cell expansion system that allows for controllable parameters. Also, cells and exosomes can be harvested at one or more time points as part of a particular regimen (WO2021263285A1).

An alternative stimulation method is patented that uses a composition containing pioglitazone, metformin, and AICAR (5-Aminoimidazole-4-carboxamide ribonucleotide). This composition will promote the production of stem cell-derived exosomes to mass-produce high-quality exosomes (KR101980453B1). 

A Chinese group has patented the treatment of stem cells with supermagnetic iron oxide-based nanoparticles for the excretion of exosomes. Following the isolation, it is suggested that the exosomes be mixed with an absorbable gel. This can further be used to promote osteogenic differentiation of stem cells or osteoblast deposition. The goal is to promote the healing of bone injuries, or osteonecrosis (CN109745341). The effect of exosomes on endothelial cells using this novel method was patented separately and was described to enhance the proliferation and migration of these cells compared to previous exosome extraction methods (CN110295142). 

Another patent focuses on the purification of exosomes derived from human urinary cells. The purification steps include a 0.22-micrometer filter membrane for the exclusion of cell debris and other impurities, followed by organelle removal via centrifugation. The patent holder claims that these exosomes can be used for treatment with the effects of resisting apoptosis, promoting angiogenesis, restoring ischemia damage, and promoting cell growth. It is further mentioned that the potential applications include bone defects, bone nonunion, femoral head necrosis, and many other application fields (CN105505854A).

A more concrete method for the application of exosomes within a hydrogel is presented in patent CN112206356. This bone-repairing hydrogel is a composite containing derivates from hydroxyapatite, hyaluronic acid, and alginic acid. The patent holder further suggests using exosomes isolated from human umbilical cord mesenchymal stem cells. The result is postulated to be an injectable bone-repairing hydrogel containing exosomes, which provides a new repair method for bone defects (CN112206356).

A different injectable hydrogel for bone injury restoration was described as temperature sensitive and contains chitosan, hydroxyapatite–collagen, and beta-sodium glycerophosphate. The hydrogel will be loaded with exosomes enriched from adipose tissue-derived stem cells, which is proposed to improve osteogenic differentiation (CN109568665). 

The following patented invention discloses a mineralized collagen gel loaded with gingival mesenchymal stem cell-derived exosomes. Next to exosomes, the gel contains mineralized collagen and chitosan/beta-sodium glycerophosphate. The exosomes are anticipated to have anti-inflammatory and osteogenesis-promoting properties. The mineralized collagen poses osteo-inductive properties, while chitosan has antibacterial activity. The resulting gel can be injected into areas with bone defects and promote local bone repair or bone amplification (CN114642630A).

An alternative, patented approach is described as a 3D bionic biological scaffold containing stem cell-derived exosomes. The scaffold is composed of methacrylic anhydride gelatin, oxidized hyaluronic acid, and modified hyaluronic acid. The scaffold mimics the acellular bone microenvironment and, in combination with exosomes, potentially promotes bone healing (CN113398332).

A multistage micro/nanostructure bone repair scaffold for freeze-drying delivery of exosomes has been patented as well. The freeze-drying delivery of exosomes into the scaffold is supposed to overcome the limited induction of osteogenesis in a regular/pure mesoporous bioactive glass scaffold. The exosomes are expected to be released slowly to meet the optimal requirements during bone repair and healing (CN112933297A).

Another patented invention describes a pedicle screw for promoting bone regeneration based on an exosome-rich, degradable hydrogel. The screw contains a hollow channel that can be filled with the exosome-containing hydrogel. After the screw is placed in the bone of a patient, the channel is loaded with hydrogel, and the exosomes will be released over a long time through the side holes of the channel. This is supposed to promote bone formation and bone marrow mesenchymal stem cell proliferation (CN113768597A).

While the previous patented inventions focused on the delivery of exosomes within gels or scaffolds, the following patents focus on other delivery systems containing exosomes to prevent or treat osteoporosis. The patented carrier preparations include physiological saline, salicylic acid, phosphate-buffered saline, starch tablets, and capsules. The targeted concentration of exosomes is 200–2000 μg/mL. The expected improvement to be achieved by this patent is increased biocompatibility and a decreased immune response (CN114246882A).

A patent from a Korean group describes the use of an extract containing exosomes from the innermost of the placenta, known as chorion. The patent protects the use of chorion extracts containing exosomes in any bone disease to promote osteogenesis (WO2022235031A1).

The isolation of exosomes from adipose tissue-derived stem cells to prevent or treat osteoporosis is patented as well. According to the presented invention, these exosomes can facilitate osteogenesis and enhance bone density (EP3659611A2).

Bone marrow-derived type M1 macrophages are cocultured with myelin sheath fragments for efficient secretion of exosomes in another patent. The isolated exosomes can be taken up by microvascular endothelial cells, which have potential applications in different organs (CN114480278).

A novel approach in which exosomes from human-induced pluripotent stem cells or embryonic stem cells are loaded with resveratrol is patented. Resveratrol is a naturally occurring polyphenol that is known to improve bone mineral density in model organisms. The patent holder claims that the biological effect of resveratrol is greatly improved by combining the agent with exosomes compared to previous delivery methods. In general, the loading of exosomes with pharmacological agents seems to improve their curative effects on bone and joint degenerative diseases (CN110151726).

An American group patented the engineering of exosomes from mesenchymal stem cells for medical purposes. The invention relates to the composition of generated exosomes and claims to result in exosomes containing increased amounts of osteo-inductive or immunomodulatory factors compared to naturally occurring exosomes (CA3106818A1).

Another patented invention describes the use of parathyroid hormone (PTH) to pre-treat bone marrow-derived mesenchymal stem cells. Exosomes derived from this treatment (ExoPTH) are proposed to have strong anti-inflammatory features and a cartilage-repairing effect in patients with osteoarthritis (CN114621918A).

Also, the use of plant-derived exosomes for the induction of chondrogenic or osteogenic differentiation of stem cells was patented in 2022. The invention is supposed to improve the rate of those differentiations and enhance the strength of the treated tissue (WO2022146374A2).

### 3.2. Clinical Studies on Exosomes in Bone Regeneration

Next to patents, the use of exosomes for therapeutic issues came into focus. An online database search (www.clinicaltrials.gov; accessed 12 October 2023) using the keyword “exosomes” resulted in 104 running trials. While seven are in early phase 1, most of the studies are currently in phase 1 and phase 2. So far, only five have reached phase 3 or phase 4, respectively (Figure 6).

When combining the terms “exosomes” and “bone”, only seven studies were listed (see Table 5). Three studies investigate the safety and efficacy of administered exosomes via injection, and three other studies use exosomes as biomarkers. In safety trials, exosomes derived from bone marrow MSC or blood (platelet-rich plasma, PRP) were reused for the treatment of acute respiratory distress syndrome (NCT05354141; NCT04493242) or low back pain (NCT04849429). The trials with a focus on exosomes as a biomarker are interestingly linked to bone metastasis (NCT03895216), osteosarcoma, or lung metastasis (NCT05101655; NCT03108677). So far, one currently running trial uses exosomes for bone grafts. In the trial, bone formation will be evaluated following the use of commercial bone substitutes with conditioned medium from adipose tissue-derived MSC containing exosomes (NCT04998058).

## 4. Discussion/Outlook

Exosomes play a major role in cell–cell communication. Therefore, it is not surprising that these secreted vesicles also contribute to angiogenesis and vascular development in bone. The angiogenic properties of mesenchymal stem cells and their exosomes are investigated by many groups with regard to their effects on endothelial cells [160,184]. The research projects mainly focus on two types of MSCs: adipose tissue-derived MSCs and bone marrow-derived MSCs. Data suggest that exosomes from both sources are taken up by endothelial cells, leading to increased proliferation, migration, and enhanced angiogenesis. On the other hand, it must be noted that some studies have shown that exosomes can also induce inflammation that can further result in endothelial dysfunction. Improved isolation and characterization methods might help to overcome these undesired side effects [185]. BM-MSC-derived exosomes seem to independently activate major endothelial signaling pathways, such as the VEGF and Hippo signaling pathways, regulate cell-to-cell contact, and regulate actin cytoskeleton dynamics [139,140,141,142,143,144,145,146]. 

The pro-angiogenic effects of MSC-derived exosomes can be further enhanced by various strategies. Preconditioning techniques such as inducing hypoxia or using chemical compounds were conducted to stimulate both exosome production and angiogenesis. Exosomes derived after such preconditioning seem to act mainly through various miRNAs as messengers, promoting angiogenesis via different pathways [147,148,149,150,151]. Genetic engineering performed to overexpress proteins or miRNAs could also boost angiogenic effects, while some genetic modifications have the opposite effect on endothelial cells [143,154]. Whereas preconditioning and engineering are promising tools, they also have disadvantages. The preconditioning of MSCs with chemicals, e.g., fails to reduce aggregation of exosomes during treatment. Moreover, the long-term effects of preconditioning on the physiological properties of MSCs require further evaluation [186,187,188]. Clearly, the engineering procedures still fail to produce the desired exosomes consistently, making the step to manufacturing adaptation difficult [188,189,190]. Physical conditions, such as culturing MSCs in a 2D or 3D environment, also had an impact on the secreted exosomes: 3D-derived exosomes caused a better outcome on proliferation, migration, tube formation, and in vivo angiogenesis in comparison to 2D-derived exosomes. Ultimately, for regenerative medicine applications like bone regeneration, exosomes were loaded onto gels or scaffolds for delivery, which also improved results since a three-dimensional structure not only better reflects the natural topology but also enables investigation for spatial and temporal effects [156]. 

Various approaches to simultaneously promote osteogenesis and angiogenesis in the context of bone repair and regeneration have been conducted as well. These studies focused mainly on the differentiation of mesenchymal stem cells into osteoblasts and their impact on endothelial cells, particularly with respect to proliferation, migration, and tube formation in vitro [165,168]. Moreover, in vivo experiments involving the use of exosomes from different sources have demonstrated improved osteogenesis and angiogenesis in both animal models and patients. The combination of exosomes and scaffolds was explored for these purposes to progress the effect on bone regeneration and blood vessel formation [169,171]. The results demonstrated enhanced bone regeneration and enhanced proliferation, migration, and angiogenic differentiation of endothelial progenitor cells (EPC) [156]. Next, in hypoxia models, various studies have been conducted to investigate the impact of exosomes on bone vascular development using disease models such as vascular dementia, diabetic bone defects, and osteoporosis, which provide another avenue for exploring bone vascular development with respect to naturally occurring deregulation and thus giving an insight into the natural processes as well as proving a target for further applications for exosomes in various diseases [176,177,179,180]. Taken together, MSC-derived exosomes could be a useful tool for therapeutic purposes in angiogenesis-driven bone repair, regeneration, and age-related defects in the future. However, so far, the lack of consensus on recently established conditions is still leading to divergence in results and desired effects. This could be a result of exosome heterogeneity and different cell sources, as well as different isolation techniques and pre-treatments.

The increasing interest in exosomes and the accumulated data thereof also led to numerous patents and clinical trials. The patents cover various methods and applications of exosomes in the context of bone regeneration and osteoporosis treatment, ranging from production and purification to innovative delivery systems and therapeutic applications (see Table 4). However, strict regulations and overall uncertainties with respect to biological and therapeutic definitions of extracellular vesicles or exosomes pose a huge challenge to the application of the above-mentioned and more advanced patents and their potential market. The novelty of this approach results so far in the focus on a safe administration of exosomes or their use as biomarkers in clinical trials (see Table 5). A much more basic investigation of exosome cargo with respect to reproducible content and its impact on recipient cells must be performed. Points that require more investigation are the isolation and purification of exosomes, the loading of therapeutics into exosomes, and the delivery of cargo to target cells. However, the current lack of standards might be the biggest task before bringing exosome-based technologies into clinics on a regular basis. The major reason for this is clearly the lack of suitable quality controls. Test systems to distinguish active extracellular particles from those that are not, or “potency assurance,” are still missing. Such an approach needs to reduce the risk of potency loss due to factors such as manufacturing control or potency lot control release to ensure that the final product has reproducibly the capacity to achieve the therapeutic effect. In addition, standards for large-scale preparation under GMP guidelines are still limited. All this must be accomplished before clinical treatment of bone repair using extracellular vesicles such as exosomes can be executed in clinical trials for the treatment of humans on a regular basis. While this presents a big challenge, the collaboration of scientists, clinicians, and biomedical technology authorities could help to overcome current limitations, even if, to some extent, an adaptation of current laws might be necessary. The complexity of cells and extracellular vesicles such as exosomes, where not only one molecule displays an effect but several components such as proteins and nucleic acids lead to an additive and may even have synergistic effects, is highly complex. Maybe even collaboration with computer experts to exert KI is needed, which already shows promising results in other complex biological and medical systems. Recently, the first exosome treatment was approved by the FDA to enter human clinical trials, giving hope that this new acellular approach, promising to circumvent at least some of the often-strict legal regulations for the use of cellular systems, is not only rapidly developing but might also overcome this barrier and cross the “Valley of Death” between research and application [191,192]. The demand is high, as is the hope. 

## Figures and Tables

**Figure 1 ijms-25-05204-f001:**
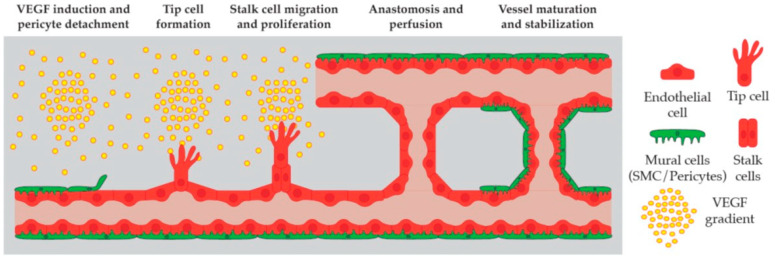
The process of angiogenesis. Proangiogenic molecules like VEGF initiate the process, and pericytes detach from the preexisting blood vessel. This is followed by the formation of a tip cell that leads the way into an avascular area and is followed by proliferative stalk cells. The newly formed vessel anastomoses with another preexisting vessel, which is followed by lumen formation and the establishment of blood flow. As a final step, the newly formed blood vessel undergoes maturation by recruiting pericytes.

**Figure 2 ijms-25-05204-f002:**
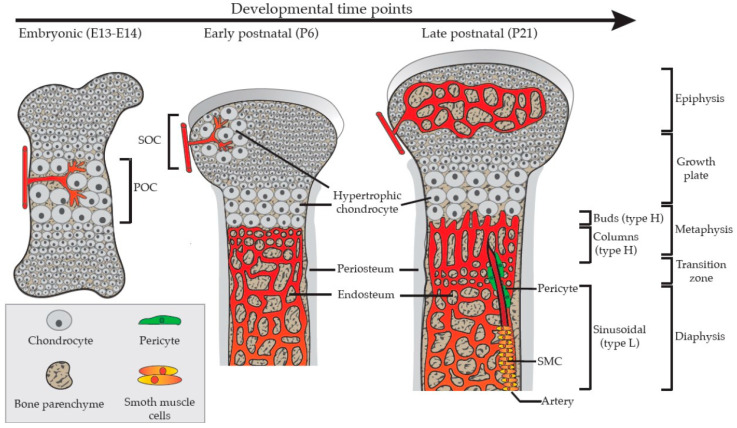
The procedure of vascular development in the bone. During mouse embryonic development (E13–E14), hypertrophic chondrocytes induce blood vessel invasion into an avascular cartilage template to form the primary ossification center (POC), coupling angiogenesis and osteogenesis. During early postnatal development (P6), a secondary ossification center (SOC) is formed due to hypertrophic cartilage at the distal end of the long bone. During late postnatal (P21) growth and bone extension, morphologically and molecularly distinct capillary populations are formed. One subpopulation is characterized by high expression of CD31 and endomucin (EMCN), called type H blood vessels. These include the buds located at the growth plate and the metaphyseal vessel columns. In contrast, the L-type sinusoidal ECs express low levels of CD31 and EMCN and are located in the bone marrow.

**Figure 3 ijms-25-05204-f003:**
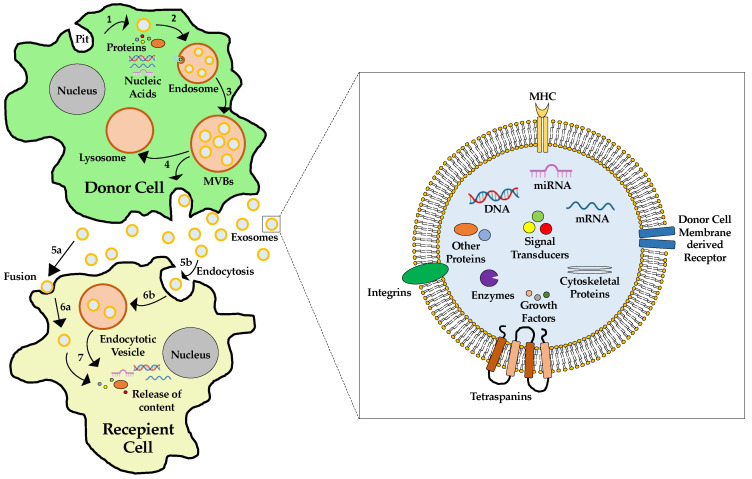
Biogenesis and content of exosomes. Left panel: After the formation of a pit (1) within the donor cell membrane, intracellular endosomes are loaded with cytosolic cargo (2). The endosomes form multi-vesicular bodies (3), which are either degraded by the lysosome (4) or exosomes and secreted into the intercellular space (4). Exosomes can be taken up by a recipient cell via fusion (5a) or endocytosis (5b). Within the recipient cell, exosomes stay within the cytosol after direct fusion with the membrane (6a), or an endocytic vesicle is formed (6b). The cargo is then released into the cytosol of the recipient cell (7). Right panel: Exosomes consist of a double membrane in which receptors and proteins of the donor cell are incorporated. The cargo of exosomes can be numerous biological compounds, including large proteins like cytoskeletal proteins, enzymes, signal transducers, growth factors, or nucleic acids like DNA, mRNA, and miRNA.

**Figure 4 ijms-25-05204-f004:**
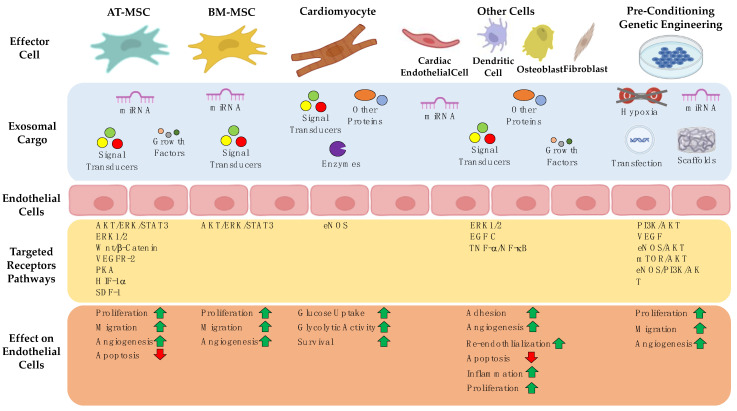
Exosomes are involved in angiogenesis. Several types of exosomes from different cell types influence endothelial cells. These exosomes can also be enriched or manipulated by genetic engineering or pre-conditioning of the effector cells. The cargo of exosomes varies from proteins to nucleic acids. Within endothelial cells, the content of exosomes, which can also be encapsulated in scaffolds, triggers different pathways. These pathways activate numerous reactions in endothelial cells, like proliferation, migration, adhesion, angiogenesis, inflammation, or apoptosis.

**Figure 5 ijms-25-05204-f005:**
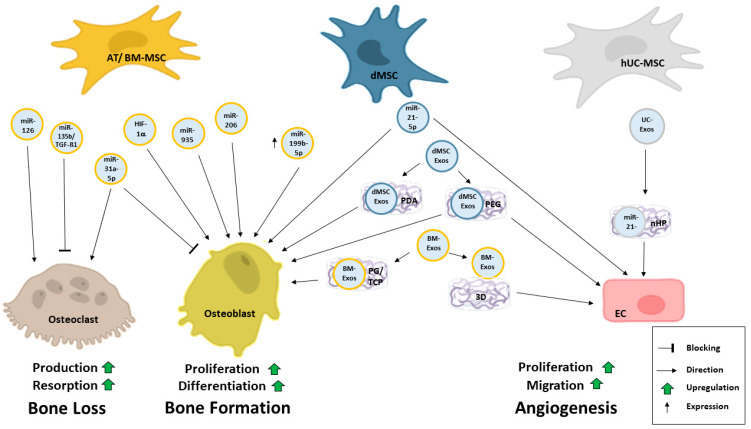
MSC-derived exosomes in bone angiogenesis. Exosomes derived from different types of MSCs have been shown to target three main cell types during bone angiogenesis. They can enhance angiogenesis via endothelial cells or improve bone formation by influencing osteoblasts. Also, bone loss is mediated by exosomes targeting osteoclasts. Several studies have shown that microRNAs (miRNAs) influence bone angiogenesis. Exosomes and their content directly affect the target or effector cells. Their exosomes can be preconditioned or embedded into scaffolds.

**Figure 6 ijms-25-05204-f006:**
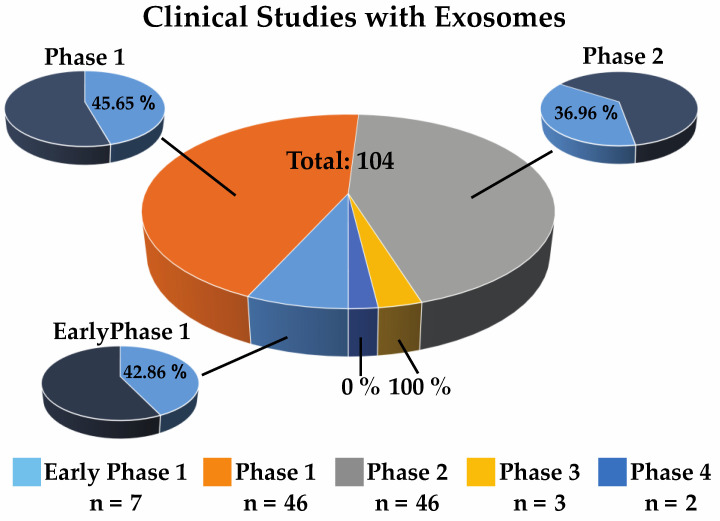
Clinical studies on exosomes. A total of 104 clinical studies were summarized at the end of 2023. The majority of the studies focus on wound healing models. The number within the legend gives the number of studies within the respective clinical phases. In early phase 1, 42.86% of the studies carry the term exosomes within their title or the description of the study. Within phase 1 studies, 45.65%, and within phase 2 studies, 36.96%, entail the term exosomes. There were three studies in phase 3 and two in phase 4.

**Table 1 ijms-25-05204-t001:** Classification of extracellular vesicles.

Type	Size [nm]	Markers	Biogenesis	Composition
Endosomal exosomes	50–100	CD63, CD9, CD81	Endosomes mature, and multivesicular bodies are formed, containing intraluminal vesicles that fuse with the plasma membrane for release	Lipid membrane
				Nucleic acids
				Proteins
				Tetraspanins
Microvesicles (MVs)	20–1000	Annexin A1, ARF6	Direct budding of the plasma membrane as a result of calcium influx and remodeling of the cortical cytoskeleton	Lipid membrane
				Nucleic acids
				Proteins
				Tetraspanins
Membrane particles	50–600	CD133+/CD63−	Direct budding of plasma membrane	Lipid membrane
				Nucleic acids
				Proteins
				Organelles
				Cytoskeleton
Apoptotic vesicles	1000–5000	Annexin V, C3b, phosphatidylserine	Cytoplasmic fragmentation during apoptosis	Lipid membrane
				Nucleic acids
				Proteins
				Apoptotic markers
				Organelles
				Fragments of nucleus

**Table 2 ijms-25-05204-t002:** Comparison of different isolation methods.

Method	Mechanism	Benefit	Disadvantage
Ultracentrifugation	Density	Well established	Time-consuming, potential damage to exosomes
Capture beads	Affinity	Highly specific and quick	Costly, loss of product due to heterogeneity, potential damage to exosomes
Magnetic-activated cell sorting (MACS)	Affinity	Highly specific and quick	Costly, loss of product due to heterogeneity, potential damage to exosomes
Precipitation (e.g., using polymers)	Solubility	Fast and affordable	Undefined efficiency
Size-exclusion chromatography	Molecular weight	Optimization process following other methods	Less efficient to LDL and other aggregates
Ultracentrifugation	Density	Well established	Time-consuming, potential damage to exosomes
Capture beads	Affinity	Highly specific and quick	Costly, loss of product due to heterogeneity, potential damage to exosomes

**Table 3 ijms-25-05204-t003:** Comparison of methods for characterization of exosomes.

Method	Principle	Advantage	Disadvantage
Atomic force microscopy (AFM)	Reflected laser beam	Presence of exosomes, size, and morphology	No quantification, potential morphological artifacts
Transmission electron microscopy (TEM)	Scattered electrons	Presence of exosomes, size, and morphology	No quantification, potential morphological artifacts
Nanoparticle tracking analyses (NTA)	Motion of suspended particles	Exosome size, distribution, and concentration	Difficult to distinguish exosomes from aggregated proteins
Tunable resistive pulse sensing (TRPS)	Changes in electrical resistance	Exosome size, distribution, and concentration	Particles might require different nano pore sizes
RNA detection	Polymerase chain reaction (PCR/qPCR)	High sensitivity and high throughput	Limited multiplex capability
RNA detection	Transcriptomics (SGS/TGS)	High sensitivity and quantitative, small sample input	Time consuming, low throughput
Protein detection	Western blotting	Detection of specific exosome subsets, cost-effective	Low sensitivity, no size information, antibody availability
Flow cytometry	Scattered light	High throughput, particle size, and number	Not suitable for very small particles (below 200 nm)

**Table 4 ijms-25-05204-t004:** Patents for exosomes in bone regeneration.

Patent Number	Publication Year	Patent Title	Subject Matter
CN105505854A	2016	Acquisition method for exosomes derived from human urinary cells and application	Exosome purification
CN110151726	2019	Use of resveratrol-loaded human pluripotent stem cell exosomes for preparing drugs for treating bone and joint degenerative diseases	Exosomes to deliver pharmaceuticals (e.g., resveratrol)
CN109745341	2019	Ferroferric oxide superparamagnetic nanoparticle stimulating stem cell for exosome osteogenesis	Increased exosome production (magnetic stimulation)
CN110295142	2019	Exosomes of bone marrow mesenchymal stem cells for promoting angiogenesis and preparation method and application thereof	Promote angiogenesis
CN109568665	2019	Temperature-sensitive injectable hydrogel loaded with adipose stem cell exosomes, and preparation method and application thereof	Novel hydrogel to deliver exosomes to bone
KR101980453B1	2019	Composition for promoting the production of stem cell-derived exosomes	Promote the production of stem cell derived exosomes
EP3659611A2	2020	Composition for preventing or treating osteoporosis containing exosomes extracted from stem cells as active ingredient	Alternative exosome source (adipose tissue-derived stem cells)
WO2020256520A1	2020	Method for producing exosomes by electrical stimulation	Increased exosome production (electrical stimulation)
CA3106818A1	2020	Engineered exosomes for medical applications	Engineered exosomes from mesenchymal stem cells
CN113398332	2021	three-dimensional bionic biological scaffold containing stem cell exosomes and application	Novel scaffold for exosome delivery (three-dimensional bionic biological scaffold)
CN112206356	2021	Injectable bone-repairing hydrogel containing exosomes of human umbilical cord mesenchymal stem cells and preparation method of injectable bone-repairing hydrogel	Novel hydrogel to deliver exosomes to bone
CN112933297A	2021	Multistage micro/nanostructure bone repair scaffold for freeze-drying delivery of exosomes	Novel scaffold for exosome delivery (multistage micro/nanostructure)
CN113768597A	2021	Pedicle screw for promoting bone regeneration based on exosome-rich degradable hydrogel	pedicle screw coating hydrogel-loaded with exosomes
WO2021263285A1	2021	Large-scale production of exosomes from primed mesenchymal stromal cells for clinical use	Increased exosome production (For large-scale production)
CN114480278	2022	High-efficiency expression method for inducing M1-type bone marrow macrophage exosomes based on myelin sheath fragments and application of high-efficiency expression method	Alternative exosome source (M1 type macrophage)
WO2022146374A2	2022	Use of plant-derived exosomes for inducing differentiation of stem cell sources into cartilage and bone cells	Alternative exosome source (plant-derived exosomes)
WO2022235031A1	2022	Pharmaceutical composition comprising chorion extract-derived exosomes as an active ingredient for promoting osteogenesis	Alternative exosome source (chorion-derived exosomes)
CN114246882A	2022	Application of mesenchymal stem cell exosomes in preparation of medicine for preventing and treating osteoporosis	Carrier preparation for exosome delivery
CN114621918A	2022	Extraction method and application of PTH-stimulated bone marrow mesenchymal stem cell exosomes	Novel exosome production (Pretreatment with PTH)
CN114642630A	2022	Mineralized collagen gel loaded with gingival mesenchymal stem cell exosome and preparation method thereof	Novel hydrogel (mineralized collagen gel) Alternative exosome source (gingival mesenchymal stem cell)

**Table 5 ijms-25-05204-t005:** Clinical trials using exosomes for disease treatment or as disease biomarkers.

Title	ID	Place	Condition
Identification and characterization of predictive factors of onset of bone metastasis in cancer patients	NCT03895216	Palermo, Italy	Bone metastases
Autogenous mesenchymal stem cell culture-derived signaling molecules as enhancers of bone formation in bone grafting	NCT04998058	Porto Alegre, Brazil	Alveolar bone loss
Construction of microfluidic exosome chip for diagnosis of lung metastasis of osteosarcoma	NCT05101655	Shanghai, China	Osteosarcoma
Circulating exosome RNA in lung metastases of primary high-grade osteosarcoma	NCT03108677	Shanghai, China	Osteosarcoma
Intra-discal injection of platelet-rich plasma (PRP) enriched with exosomes in chronic low back pain	NCT04849429	Rudrapur, India	Chronic low back pain
Extracellular vesicle treatment for acute respiratory distress syndrome (ARDS)	NCT05354141	Chandler, USA	Acute respiratory distress syndrome (ARDS)
Extracellular vesicle infusion treatment for COVID-19 associated ARDS (EXIT-COVID19)	NCT04493242	Sheffield, USA	COVID-19, ARDS

## Data Availability

Not applicable.

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
