# Peer review of "Exosomes: A New Hope for Angiogenesis-Mediated Bone Regeneration"

_ijms, 2024, doi:10.3390/ijms25105204_

Round 1
Reviewer 1 Report
Comments and Suggestions for Authors
The review discusses the role of exosomes in bone vascular development and homeostasis, including a discussion on patented ap-18 plications and an update on ongoing clinical trials using exosome for bone regeneration. In general, it is well written and well structured.
1. While the research on exosomes is promising, there are still challenges to overcome in terms of producing them on a large scale for clinical applications. It would be beneficial to discuss any limitations or challenges in the current understanding/application of exosome and the translation of these findings into clinical practice.
2. The inclusion and exclusion criteria for previous studies are not well-defined. Not sure if this review effectively covered the current state of research?
3. This review doesn’t include the discussion/analysis of collected studies.
Author Response
Dear colleagues,
We appreciate the time and effort you and each of the reviewers have dedicated to refine our paper. Thank you.
We have edited the manuscript addressing the suggestions of the reviewers for improvements. The revisions are highlighted and can be followed by Microsoft Word “Track Changes” function. To facilitate your review of our revisions, the following is a point-by-point response to the questions and comments of the reviewers.
Reviewer 1:
The review discusses the role of exosomes in bone vascular development and homeostasis, including a discussion on patented ap-18 plications and an update on ongoing clinical trials using exosome for bone regeneration. In general, it is well written and well structured.
- While the research on exosomes is promising, there are still challenges to overcome in terms of producing them on a large scale for clinical applications. It would be beneficial to discuss any limitations or challenges in the current understanding/application of exosome and the translation of these findings into clinical practice.
We absolutely agree, that there are still major challenges to overcome before we shall see reasonable numbers of clinical applications. Therefore we have rewritten discussion and outlook putting more emphasis on the limitations and challenges. Thank you for the comment.
- The inclusion and exclusion criteria for previous studies are not well-defined. Not sure if this review effectively covered the current state of research?
For patents an online database research using WIPO Patenscope (https://www.wipo.int/portal/en/) and Espacenet (https://worldwide.espacenet.com) was performed using the terms “exosomes” and “bone”.
For the clinical trials an online database research (www.clinicaltrials.gov; accessed October, 12th, 2023) using the keyword “exosomes”. Also combined search terms “exosomes” and “bone” were applied.
Thank you for the helpful comment. It slipped our notice. We added 24 recent publication that fit this review throughout the manuscript. In detail, we added 4 publication from 2022, 9 from 2023, and 10 from 2024.
- This review doesn’t include the discussion/analysis of collected studies.
An additional column was added to table 1 (now table 4) summarizing the subject matter of each patent. In addition a more thorough evaluation of the collected studies was added in discussion/outlook.

Reviewer 2 Report
Comments and Suggestions for Authors
REVIEWER COMMENTS
(MAJOR COMMENTS)
This research article "The Role of Exosomes in Bone Specific Angiogenesis and Vascular Homeostasis" With Manuscript ID: ijms-2959607. The major comments are mentioned below,
Comment 1: Tile needs to be revised as it is too general.
Comment 2: The abstract and keywords need to be revised for better clarity of study.
Comment 3: Points 1.1, 1.2, and 1.3 should be very precise, reduce the length.
Comment 4: In point 1.4 include a recent case study.
Comment 5: In point 1.5, classification of exosomes and isolation techniques should be explained in detail.
Comment 6: In point 2 2. Exosomes in Bone Vascular Development and Homeostasis, add 2 to 3 recent case studies to support the content.
Comment 7: In point 2.1 Exosomes in Angiogenesis, add 2 to 3 case studies.
Comment 8: Arrange data in ascending order of year. Add inference of the patent in Table 1.
Comment 9: The resolution of Figure 6 needs to improve.
Comment 10. Add 1 more figure and 2 tables to match journal standards.
Comment 10: Recent references are missing particularly 2022, 2023.
Comment 11: Grammar and language corrections are needed throughout the manuscript.
Comments on the Quality of English LanguageNeed improvment.
Author Response
Dear colleagues,
We appreciate the time and effort you and each of the reviewers have dedicated to refine our paper. Thank you.
We have edited the manuscript addressing the suggestions of the reviewers for improvements. The revisions are highlighted and can be followed by Microsoft Word “Track Changes” function. To facilitate your review of our revisions, the following is a point-by-point response to the questions and comments of the reviewers.
Reviewer 2:
This research article "The Role of Exosomes in Bone Specific Angiogenesis and Vascular Homeostasis" With Manuscript ID: ijms-2959607. The major comments are mentioned below,
- Tile needs to be revised as it is too general.
We changed the title as follows:
Exosomes: A new hope for angiogenesis mediated bone regeneration
- The abstract and keywords need to be revised for better clarity of study.
We modified abstract and keywords.
- Points 1.1, 1.2, and 1.3 should be very precise, reduce the length.
1.1 Angiogenesis was reduced from 487 words to 380 words.
- Bone Specific Vascular Development was reduced from 249 to 228 words. This was already a short chapter.
- Vascular Homeostasis in Bone was reduced from 641 to 545 words
- In point 1.4 include a recent case study.
We used PubMed restricted to the last five years and the search terms case reports (36.000 pages), case reports + osteoporosis (100 pages), case reports + exsosome (100 pages), case reports + osteoporosis + exsosome (0 pages), case reports + bone (1 page, four papers, none of which fit this review). When using only the search term bone vascular homeostasis, three publications could be found. Unfortunately, none of these three case reports fit to the content of this review and thus were included. Thus, we have added a recent clinical trial on postmenopausal women with low BMD or osteoporosis.
- In point 1.5, classification of exosomes and isolation techniques should be explained in detail.
In line with the suggestion to add more tables, we summarized classification, isolation techniques and characterization methods each within a new table:
Classification of cellular vesicles is now explained more detailed in table 1.
Different isolation techniques are now compared more detailed in table 2.
Characterization methods are now compared more detailed in table 3.
- In point 2 2. Exosomes in Bone Vascular Development and Homeostasis, add 2 to 3 recent case studies to support the content.
We used PubMed restricted to the last five years and the search terms case reports MSC exosome however even without the keyword vascular development this resulted in no match. However, we will be happy to consider suggestions of case studies from the reviewer for the suggested topic.
- In point 2.1 Exosomes in Angiogenesis, add 2 to 3 case studies.
This is now the third topic were the reviewer asks to add case studies. We find this quite unusual. While there is no doubt that case studies might be interesting, especially for MDs, as they can point to rare occurrences, they are usual individual cases, which might be of minor interest for common reader in a more general review. Also case studies are found quite often as the topic in bachelor or master theses to develop a qualitative empirical research approach, lacking the quantitative analysis and thus are not published in a peer-reviewed journal. In line with this, we have been unable to find case studies which match the three topic “exosome in angiogenesis” as was requested. As before we are happy to consider suggestions from the reviewer.
- Arrange data in ascending order of year. Add inference of the patent in Table 1.
The order of the table content was in line with the appearance in the text. It is now in ascending order of year. An additional column was added to table 1 (now table 4) summarizing the subject matter of each patent.
- The resolution of Figure 6 needs to improve.
The resolution and overall appearance of figure 6 was improved.
- Add 1 more figure and 2 tables to match journal standards.
We find this comment quite astonishing.
Our review had 6 figures and in addition 2 tables, all made solely for this review. Also its goes without saying that there can be figures and tables with are elaborate, taking a lot of research and time, and others are just reproduced from previous journals, or are quite simple, such as just giving a structural formula. Both, of course, might be supporting the text. However, creating a new, summarizing figure from a large number of publications can have additional benefits. Our figures and tables clearly belong in this group.
While we feel that not just counting the number of figures and tables sets the standard of a journal, we also want to point out that we clearly meet the current numbers usual in this particular journal. To do so we checked 10 reviews in this journal - not picking those in favor of our arguing, but just looking at the last ten published. We found the following result:
Int. J. Mol. Sci. 2024, 25(7), 4102; https://doi.org/10.3390/ijms25074102 (registering DOI): 4 figures and 1 table
Int. J. Mol. Sci. 2024, 25(7), 4101; https://doi.org/10.3390/ijms25074101 (registering DOI): 2 figures and 0 tables
Int. J. Mol. Sci. 2024, 25(7), 4100; https://doi.org/10.3390/ijms25074100 (registering DOI): 2 figures and 5 tables
Int. J. Mol. Sci. 2024, 25(7), 4099; https://doi.org/10.3390/ijms25074099 (registering DOI): 5 figures and 2 tables
Int. J. Mol. Sci. 2024, 25(7), 4096; https://doi.org/10.3390/ijms25074096 (registering DOI): 3 figures and 5 tables
Int. J. Mol. Sci. 2024, 25(7), 4102; https://doi.org/10.3390/ijms25074102 (registering DOI): 4 figures and 1 table
Int. J. Mol. Sci. 2024, 25(7), 4092; https://doi.org/10.3390/ijms25074092 (registering DOI): 4 figures and 1 table
Int. J. Mol. Sci. 2024, 25(7), 4057; https://doi.org/10.3390/ijms25074057:
2 figures and 3 tables
Int. J. Mol. Sci. 2024, 25(7), 4050; https://doi.org/10.3390/ijms25074050:
4 figures and 3 tables
Int. J. Mol. Sci. 2024, 25(7), 4051; https://doi.org/10.3390/ijms25074051:
3 figures and 0 tables
As can be seen, there is not a single recent publication, which is even matching our number of figures. Six of the ten publications have the same or less tables than we had, too. Five of them are having half our number of tables or less. When summarizing figures and tables, none of them has a total of 8, as we have. Only the one with the highest number is matching our numbers.
Thus, I cannot help to wonder what to make out of such a comment. I am simply unable to follow the argumentation. Nevertheless, we added three additional tables, having now a total of 11 figures/tables to meet the expectation of the reviewer.
10(b?). Recent references are missing particularly 2022, 2023.
Thank you for the helpful comment. We added 24 recent publication that fit this review throughout the manuscript. In detail, we added 4 publication from 2022, 9 from 2023, and 10 from 2024.
- Grammar and language corrections are needed throughout the manuscript.
We assume this comment is mainly based on the titles of patents and clinical studies, which are indeed displaying an unusual language/grammar. However, we have to give the titles as published. Additionally we asked a native speaker for corrections of the whole manuscript.

Round 2
Reviewer 2 Report
Comments and Suggestions for Authors
I am satisfied with the revision corrections and effort taken by the authors to improve the standard and quality of the article in order to match the journal standards.